# Dynamic Balancing of Humanoid Robot with Proprioceptive Actuation: Systematic Design of Algorithm, Software, and Hardware

**DOI:** 10.3390/mi13091458

**Published:** 2022-09-02

**Authors:** Yan Xie, Jiajun Wang, Hao Dong, Xiaoyu Ren, Liqun Huang, Mingguo Zhao

**Affiliations:** 1Beijing Research Institute of UBTECH Robotics, Beijing 100084, China; 2Department of Automation, Tsinghua University, Beijing 100084, China; 3Beijing Innovation Center for Future Chips, Beijing 100084, China

**Keywords:** whole-body control, hierarchical optimization, humanoid robot balance, proprioceptive actuation

## Abstract

For humanoid robots, maintaining a dynamic balance against uncertain disturbance is crucial, and this function can be achieved by coordinating the whole body to perform multiple tasks simultaneously. Researchers generally accept hierarchical whole-body control (WBC) to address this function. Although experts can build feasible hierarchies using prior knowledge, real-time WBC is still challenging because it often requires a quadratic program with multiple inequality constraints. In addition, the torque tracking performance of the WBC algorithm will be affected by uncertain factors such as joint friction for a large transmission ratio proprioceptive-actuated robot. Therefore, the balance control of physical robots requires a systematic solution. In this study, a robot control system with high computing power and real-time communication ability, UBTMaster, is implemented to achieve a reduced WBC in real time. Based on these, a whole-body control scheme based on task priority for the dynamic balance of humanoid robots is implemented. After realizing the joint friction model identification, finally, a variety of balancing scenarios are tested on the Walker3 humanoid robot driven by the proprioceptive actuators to verify the effectiveness of the proposed scheme. The Walker3 robot exhibits excellent balance when multiple external disturbances occur simultaneously. For example, the two feet of the robot are subjected to tilt and displacement perturbations, respectively, while the torso is subjected to external shocks simultaneously. The experimental results show that the dynamic balance of the robot under multiple external disturbances can be achieved by using strictly hierarchical real-time WBC with a systematic design.

## 1. Introduction

Scholars have studied humanoid robots for decades and have made significant progress in robot mobility, dexterity, and intelligence [1,2,3]. However, making humanoid robots work or interact with humans in a human-friendly environment faces substantial challenges. First, maintaining balance is one of the most fundamental skills of legged robots.

Due to the real-time requirements of the balance algorithm, scholars widely used the balance control method based on a simplified model in their early work. They simplified the robot into a single rigid body mounted on the top of the inverted pendulum, and the control target was its center of mass [4]. Nechev et al. [5] proposed a three-link planar model, which can include more information. Because the model’s accuracy is improved, these methods, including ankle or hip joint, will be more feasible for a physical robot. However, due to the failure to consider the motion of all joints, they will still give unnatural solutions in some cases.

Whole-body control (WBC) has been paid more and more attention by robotics in the past two decades because it makes full use of the redundant degrees of freedom of robots to complete multiple tasks simultaneously. For example, the humanoid robot has various degrees of freedom to realize balance as a high-priority task in the WBC framework. Therefore, WBC has gradually become a basic control scheme for the humanoid robot. Dietrich et al. [6] have categorized the existing WBC methods into: null-space projection-based WBC (NSP-WBC), weighted quadratic program-based WBC (WQP-WBC), and hierarchical quadratic program-based WBC (HQP-WBC).

In redundant manipulator control, the NSP-WBC realizes the task hierarchy through null-space projection [7]. Kajita et al. [8] extended this idea to the walking control of the HRP-2 robot for the first time. However, this method is limited because it cannot consider inequality dynamic constraints such as joint torque limitations, which are highly necessary to the robot’s safety.

The WQP-WBC handles this problem by formulating the tasks and constraints as a quadratic optimization problem. As a result, it can find the optimal solution that minimizes the task errors while satisfying the constraint conditions. This method has been applied to the Atlas robot to execute multi tasks during the DARPA robotics challenge [9,10,11]. However, the WQP-WBC cannot guarantee a strict task hierarchy. The soft hierarchy realized by tuning the task weight becomes a limitation when the tasks conflict.

The HQP-WBC has been devised to combine the task hierarchy and inequality constraints together [12,13,14]. The basic idea is to construct each layer as a quadratic optimization problem and solve them in a sequence where the lower-priority QPs cannot disturb the higher-priority QPs. As a result, the inequality constraints in the QP stack gradually as the priority decreases, thus leading to a time-consuming problem. Ref. [15] proposed an efficient way by introducing null-space projection to reduce the computational cost and implemented it on a torque-controlled robot Sarcos with a 1 KHz control loop. Similarly, ref. [16] implemented this method on a quadruped robot ANYmal showing a natural adaption to the terrain while walking. In short, the HQP-WBC has been a standard whole-body motion generation tool for torque-controlled robots.

In recent years, proprioceptive actuation has become a mature technology widely used in scenarios requiring back-driveability, such as legged robots. Engineers can compensate for the torque loss in the reducer online through the predetermined friction model and online parameter identification, such as the high dynamic locomotion of the MIT cheetah robot [17]. However, when the gear reduction ratio increases, the friction model becomes more complex and has stronger nonlinearity and uncertainty. Therefore, more comprehensive offline modeling and parameter identification processes, and even an online identification process, are required.

Besides the high-bandwidth characteristic of actuators, the overall performance of the robot highly depends on the control frequency. For example, ref. [18] has noticed a phenomenon in the torque-controlled robot, Mercury, that increasing the control frequency from 1 kHz to 1.5 kHz will provide a more significant posture and foot position control bandwidth. This operation puts forward a higher demand for the real-time computation of WBC. Moreover, a real-time WBC is preferred as other time-consuming techniques such as model predictive control (MPC) is usually embedded into the control framework.

This paper focuses on the dynamic balance control of a humanoid robot with proprioception actuators through algorithm, software, and hardware system integration. We first customize a prioritized hierarchy of tasks and constraints for rejecting multi-disturbances. Then, a reduced whole-body control is implemented in real-time by UBTMaster, a control system designed to provide computationally efficient WBC software and powerful computing hardware. This unique real-time computing system is not available in other robotic systems, which ensures the effective implementation of the control algorithm. Next, the model identification process covers the joint friction and model inaccuracy issues. Finally, plenty of experiments on various balancing scenarios are implemented on a robot Walker3 with proprioceptive actuation, and the performance is discussed.

## 2. Control Approach

Walker3 is a humanoid robot with two legs and a torso, as shown in Figure 1. The robot has a height of 1.6 m and weighs 43 kg. An inertial measurement unit (IMU) is mounted on the torso for state estimation, and two six-axis force sensors are installed on the soles to measure each foot’s center of pressure (CoP). Each leg has six electrical motors in series. The motors use a gear reducer to enlarge the output torque, and the gear reduction ratio ranges from 50 to 100. The actuators are controlled in real time using the EtherCAT communication protocol.

An online task planner is proposed to adjust the task trajectory. The basic idea of this planner is to ensure the foot ZMP resides in the safe region as much as possible. Then, the desired CoM trajectory is tracked by a reduced whole-body controller coupled with a hierarchical optimization solver. The hierarchy of tasks and constraints is divided into four layers according to their priorities. A quadratic optimization solves each layer, and the null-space projection guarantees the strict hierarchy among layers. After solving a sequence of QPs, the whole-body controller outputs the optimized joint torques.

Joint torques are turned into the current commands for a robot with proprioceptive actuation. The desired joint velocity commands obtained through the numerical integration of desired joint accelerations are also considered here to improve the performance of the joint-level control. The whole control architecture is illustrated in Figure 2. The kinematics solver estimates the robot’s state with the sensory data of joints and IMU.

## 3. Tasks and Constraints in Dynamic Balancing

### 3.1. Task Planner

In dynamic environments, the robot needs to properly tune the desired motion. The task planner presented here can make real-time adjustments to the task trajectory according to the ZMP state.

Unlike the method in [15], we have no extra sensors to obtain information on the moving support, apart from the force sensors to detect the contact between the foot and support. When a robot is stably resting on a moving support, the ZMP of each foot must reside inside its support polygon. Therefore, the desired motion of the foot can be set using the following rule: if the measured ZMP is inside the safe region of the support polygon, we adjust the desired position, posture, and velocity of the foot to its current state $r_{F,ref}=r_{F}$ and $v_{F,ref}=v_{F}$.

In addition, the desired motion of CoM should be adjusted along with the foot desired motion. The desired horizontal position is located in the middle of two feet:(1)$$r_{\mathrm{CoM},ref}\left(x,y\right)=\left(r_{LF,ref}\left(x,y\right)+r_{RF,ref}\left(x,y\right)\right)/2$$
The desired vertical position is set to
(2)$$r_{\mathrm{CoM},ref}\left(z\right)=\left(r_{LF,ref}\left(z\right)+r_{RF,ref}\left(z\right)\right)/2+C$$
where C is a constant value depending on the robot stand pose. The desired velocity is set to the average velocity of two feet
(3)$$v_{\mathrm{CoM},ref}=\left(v_{LF,ref}+v_{RF,ref}\right)/2$$

In our planner, the desired motion of the torso and foot contact force remains unchanged throughout the balance control. The desired motion of torso is set to $r_{T,ref}=0$, $v_{T,ref}=0$ and the desired vertical contact force is set to $F_{LF,ref}\left(z\right)=F_{RF,ref}\left(z\right)=mg/2$, where *m* is the total mass of the robot.

### 3.2. Tasks and Constraints Hierarchy

When multi-tasks have to be performed simultaneously, handling the conflicts among these objectives is crucial. The prioritized hierarchy strategy has been widely adopted in redundant robots. Motion solvers will accomplish the lower-priority tasks under the prerequisite that the higher-priority tasks are implemented first. For example, balance is always considered a top-layer priority task for a humanoid robot. As a result, the robot tends to sacrifice its posture under disturbance to ensure the feet fully contact the ground. Likewise, physical constraints concerning humanoid robot safety should be the highest priority. Table 1 specifies the hierarchy of tasks and constraints.

#### 3.2.1. Floating Base Dynamics

Humanoid robots, typical floating-based systems, are an example of underactuated systems due to their partial actuation when interacting with the environment. The configuration of a humanoid robot is represented by generalized coordinates $q={\left[\begin{array}{cc} q_{f}^{T} & q_{a}^{T} \end{array}\right]}^{T}$, $q_{f}$ represents the position and orientation of the robot free-floating body and $q_{a}$ represents the *n*-actuated joints of the robot. When the robot is in contact with the environment, the dynamic equation of the system can be fully described by
(4)$$S_{f}M\ddot{q}+S_{f}C+S_{f}G=S_{f}J^{T}F$$
where *M* is the generalized inertia matrix, *C* is the nonlinear vector including Coriolis and centrifugal forces, *G* is the gravity vector. *F* is the contact force vector, and *J* is the Jacobian matrix of the contact point. $S_{f}=\left[\begin{array}{cc} I & 0 \end{array}\right]$ is a matrix selecting the free-floating joints. Thus, actuated torque vector τ is eliminated from the dynamic equation. Instead, choosing a different selection matrix $S_{a}=\left[\begin{array}{cc} 0 & I \end{array}\right]$ will derive a linear function between τ and $\ddot{q}$, *F*. Due to such linear dependence, the whole-body dynamics of n+6 dimensions are simplified as the floating base dynamics of six dimensions. Adopting floating base dynamics is crucial to reduce the optimization time and help implement the 1KHz control loop.

The dynamic equation is essential for a physical multi-rigid-body system as the highest-priority task. Once the equation holds, the movements of the system are physically feasible. Following the dynamics equation, the second-priority task is the foot position and posture control. A good task control performance will guarantee good contact between foot and support, which is a premise for the contact force. Linear momentum control, the third priority task, has been proven to be essential for a good balance by regulating the state of CoM [19]. Finally, in the lowest-priority task, we prefer to have the torso posture control and foot contact force control on the same level. This is because we have only 30 optimization variables (including 18 for $\ddot{q}$ and 12 for *F*). While there are 6, 12, and 3 variables for dynamic equation, foot position, posture task, and linear momentum task, respectively, only 9 free variables are left for the torso posture and foot contact force control.

#### 3.2.2. Operational Space Tasks

Operational space tasks such as foot position and posture control, linear momentum control, and torso posture control can be phrased as:(5)$$J\ddot{q}+\dot{J}\dot{q}=a$$
where *J* is the Jacobian matrix of a specific task, and *a* is the task desired acceleration which can be determined by a feedforward and feedback control law. The $\dot{J}\dot{q}$ term is related to the robot state. In particular, the Jacobian matrix of the linear momentum task is also called the centroidal momentum matrix. It can be calculated using an efficient O(*n*) algorithm based on the generalized inertia matrix *M* [20].

#### 3.2.3. Robot Safety Constraints

Given the physical limitations of the robot, several safety issues must be appropriately concerned. The joint torque saturation constraint $\tau_{\min}\leq \tau \leq \tau_{\max}$ is especially important for generating control commands that are valid on a robot.

A stable contact between foot and support is an essential precondition for generating a six-dimensional contact force vector, which means the foot cannot tilt or slide relative to the support. Stable contact can be ensured from two aspects. One is the center of pressure constraint. The center of pressure at each foot must not exceed the foot’s support polygon boundary. The other is the friction cone constraint, which requires that the foot contact force stays inside the friction cones. The cones are approximated as pyramids here, so the constraint can be expressed as linear inequality.

## 4. Real-Time WBC

### 4.1. Reduced Hierarchal Whole-Body Control

The tasks can be formulated as equalities, and constraints can be formulated as inequalities. Therefore, the tasks and constraints in the same level can be stacked vertically into the form.
(6)$$\left\{\begin{array}{c} A_{i}x-b_{i}=0\\ D_{i}x-f_{i}\leq 0 \end{array}\right.$$
where $A_{i}$ is the *i*th task matrix, $b_{i}$ is the *i*th task reference vector, $D_{i}$ is the *i*th constraint matrix, $f_{i}$ is the *i*th constraint boundary vector, and $x={\left[\begin{array}{cc} {\ddot{q}}^{T} & F^{T} \end{array}\right]}^{T}$ is the optimal variables. The goal of this level is to find $\ddot{q}$ and *F* that satisfy these objectives as well as possible. The solution under such a linear inequality constraint can be solved through quadratic optimization. The tasks and constraints in different levels need to be optimized in a strict prioritized order. Solving level *p* yields an optimal solution $x_{p}^{*}$. In order to ensure the strict prioritization of tasks, the solution of level p+1 can be found in the null space of all higher-priority tasks $N_{p}^{}=N_{p-1}^{}\left(I-{\hat{A}}_{p}^{\#}{\hat{A}}_{p}\right)$. $N_{p-1}^{}$ is the null space of all tasks from level 1 to p−1. $\left(I-{\hat{A}}_{p}^{\#}{\hat{A}}_{p}\right)$ is the null space of task in level *p*. ${\hat{A}}_{p}=A_{p}N_{p-1}$ describes the task matrix of level *p* projected into the null space of all higher-priority tasks. The solution of level p+1 can be expressed as $x_{p+1}=x_{p}^{*}+N_{p}u_{p+1}$, where $u_{p+1}$ is an arbitrary vector lying in the row space of $N_{p}$. Substituting $x_{p+1}$ into the QP problem in level p+1 yields
(7)$$\begin{array}{c} \underset{u_{p+1}}{\min.}\;\;\;\;\;\;\;{∥A_{p+1}\left(x_{p}^{*}+N_{p}u_{p+1}\right)-b_{p+1}∥}^{2}\\ \;\;\;s.t.\;\;\;\;\;\;\;\;\;\;\;\;\;D_{p+1}\left(x_{p}^{*}+N_{p}u_{p+1}\right)-f_{p+1}\leq 0\\ \;\;\;\;\;\;\;\;\;\;\;\;\;\;\;\;\;\;\;\;\;\;D_{p}\left(x_{p}^{*}+N_{p}u_{p+1}\right)-f_{p}\leq 0\\ \;\;\;\;\;\;\;\;\;\;\;\;\;\;\;\;\;\;\;\;\;\;\vdots \\ \;\;\;\;\;\;\;\;\;\;\;\;\;\;\;\;\;\;\;\;\;\;D_{1}\left(x_{1}^{*}+N_{p}u_{p+1}\right)-f_{1}\leq 0 \end{array}$$

All the higher-priority constraints are stacked into the optimization to ensure the strict prioritization of constraints. Then, the recursive algorithm is used to solve the QP of each layer according to the priority order.

The slack variables are introduced initially to turn the hard constraint into a soft one. In our case, however, we notice that the optimized slack variables are always zero, which means that the solver can find the optimal result without violating the hard constraints. Therefore, the slacks are excluded from the optimization variables to reduce the computational complexity. As a result, the slack variables are omitted in the optimization problem, different from the general formulation in [15].

WBC software is developed based on C++ to implement the above algorithm effectively. Figure 3 depicts the architecture of the WBC software. The software contains four basic classes: RobotDynamics, Task, Constraint, and Wbc. These classes provide the basic interfaces for user development, and the derived classes of Walker3 are developed in this software. The following part will describe these classes in detail.

The RobotDynamics class contains the member variables related to the kinematics and dynamics of a robot, such as the number of the generalized joints, contact forces, inertia matrices, Coriolis and centrifugal vectors, gravity vectors, selection matrices, Jacobian matrices, etc.

For the Walker3 robot, it is implemented by a subclass named RobotDynamics_Walker3. The model structure of Walker3 can be constructed in the subclass directly or loaded from URDF files. Calling the calcWbcDependence()function can obtain all the required kinematics and dynamics parameters. The open-source rigid body dynamics library (RBDL), a highly efficient C++ library with some essential rigid body dynamics algorithms [21], is used here. The task and constraint classes are constructed according to Equation (Equation 6) with their member variables, including the task’s or constraint’s name, priority, dimension, matrix, reference vector (boundary vector), and the DoF of variables. Each task or constraint of Walker3 is implemented by a subclass, thus forming a task or constraint library. Using the update (const RobotDynamics &) function will update the member variables with calculated kinematics and dynamics parameters.

The Wbc class, the software’s core, contains the pointers of the other three types. It can manage the tasks’ addition, deletion, and adjustment operations and constraints. In its implementation, two subclasses based on different algorithms are developed here. One is named WqpWbc, which forms all the tasks and constraints as one quadratic optimization problem [10]. The other is HqpWbc, which implements the hierarchical quadratic optimization mentioned before.

This software avoids lots of redundant codes and improves efficiency development. Meanwhile, developers also build the dynamic model of some robots and their corresponding tasks’ and constraints library. Users can also develop their robots without rewriting the WBC solver code.

### 4.2. High-Performance Master Control System

The WBC software is embedded in a modular master control system named UBTMaster, as shown in Figure 4a. It is designed with the characteristics of real-time solid computation, extensible computing capability, and a configurable interface. As a result, it can realize the different combination configurations of typical hardware platforms, such as ARM, GPU, X86, and DSP, and expand the computing capability in different scenarios. For example, the X86 basic edition can perform 100 GFLOPS per second with the Intel Core i7-7600U.

The software architecture is illustrated in Figure 4b. The real-time operating system based on the PREEMPT_RT kernel serves real-time applications that process data as it comes in, typically without buffer delays. It ensures that the application’s task must be carried out within the defined time constraints. In the real-time communication layer, we apply the high-speed real-time bus communication protocol EtherCAT for short data update times (also called cycle times; ≤100 us) with low communication jitter (for precise synchronization purposes; ≤1 us). These two aspects can control the time jitter on a microsecond level.

In the upper layer, the roboCore runs in real time and isolates the applications from the hardware platform. As a result, users can develop their applications to meet specific requirements.

## 5. Proprioceptive Actuation with a Big Reduction Ratio

### 5.1. Joint-Level Control

Given the inputs, WBC software will output the optimized joint torque. However, the lack of a torque sensor in proprioceptive actuation does not allow direct torque control. A common workaround for this problem is to utilize an admittance coupling to convert joint torque to joint velocity [22]. However, considering that the bandwidth of admittance control will limit the torque tracking performance, we utilize direct current control.

The joint current can be approximated as a linear function of joint torque due to the negligible torque loss in the reducer for proprioceptive actuation with a small reduction ratio. However, the reducer has significant static friction with a large reduction ratio. This stiction translates into joint torque stiction of up to 5 Nm. Consequently, joint friction torque compensation is essential. Moreover, the joint velocity obtained by integrating the optimized joint acceleration can also be added to the current command as a kinematic compensation term to improve the joint impedance [11].

In this research, the final control law for the joint current is calculated as:(8)$$i^{\mathrm{cmd}}=k_{i}\left(\tau^{\mathrm{opt}}+k_{f}\tau^{f}+\tau^{\dot{q}}\right)$$
where $\tau^{\mathrm{opt}}$ is the optimized joint torque, $\tau^{f}$ is the joint friction compensation torque, $k_{f}$ is the corresponding friction compensation coefficient, $\tau^{\dot{q}}$ is the joint kinematic compensation torque. $\tau^{\mathrm{opt}}$,$\tau^{f}$, and $\tau^{\dot{q}}$ can be expressed as:(9)$$\tau^{\mathrm{opt}}=S_{a}^{}M{\ddot{q}}^{\mathrm{opt}}+S_{a}^{}\left(C+G\right)-S_{a}^{}J^{T}F^{\mathrm{opt}}$$
(10)$$\tau^{f}=\left\{\begin{array}{c} F_{c}+F_{v}\left(\int {\ddot{q}}^{\mathrm{opt}}dt-{\dot{q}}^{*}\right),\;\;\;\int {\ddot{q}}^{\mathrm{opt}}dt\geq {\dot{q}}^{*}\\ \int {\ddot{q}}^{\mathrm{opt}}dt\frac{F_{c}}{q^{*}},\;\;\;\;\;-{\dot{q}}^{*}\leq \int {\ddot{q}}^{\mathrm{opt}}dt\leq {\dot{q}}^{*}\\ -F_{c}+F_{v}\left(\int {\ddot{q}}^{\mathrm{opt}}dt+{\dot{q}}^{*}\right),\;\;\;\int {\ddot{q}}^{\mathrm{opt}}dt\leq -{\dot{q}}^{*} \end{array}\right.$$
(11)$$\tau^{\dot{q}}=k_{\dot{q}}\left(\int {\ddot{q}}^{\mathrm{opt}}dt-\dot{q}\right)$$
where $\tau^{\mathrm{opt}}$ is reorganized from the full dynamics. The joint friction compensation torque $\tau^{f}$ is modeled as two parts: Coulomb and viscous friction. $F_{c}$ is the Coulomb friction and $F_{v}$ is the viscous friction coefficient. ${\dot{q}}^{*}$ is a user-defined value to prevent a sudden jump in joint friction compensation torque as the motor rotates reversely. $k_{\dot{q}}$ is a gain acting on the difference between the desired joint velocity $\int {\ddot{q}}^{\mathrm{opt}}dt$ and the measured joint velocity $\dot{q}$.

### 5.2. Model Identification

Model identification is an effective method used for obtaining a robot’s dynamic parameters. Besides the concerned dynamic parameters such as links’ mass, inertia, and center of mass, the joint friction model can also be incorporated into the linearized dynamic equation [23]. Here, the Coulomb–viscous friction model is preferred due to its linear expression.

The main process can be divided into three parts.

#### 5.2.1. Linearization of Dynamic Equation

The recursive Newton-Euler equation is used to reorganize the joint torque τ as a linear function of dynamic parameters π (including joint friction parameters), given that τ=Yπ. *Y* is the identification matrix and can be uniquely determined by joint motion *q*, $\dot{q}$, and $\ddot{q}$.

#### 5.2.2. Optimal Excitation Trajectory

The excitation trajectory is parameterized first by a finite Fourier series function and then optimized for the minimum of a user-defined cost function [24] while satisfying the constraint conditions. The series and base frequency in the Fourier series are set at 5 and 0.1 Hz, respectively. The condition number of the Y matrix is closely relative to the mean square error of identification results and thus selected as the cost function.

#### 5.2.3. Dynamic Parameters Optimization

An optimization problem is constructed to find optimal dynamic parameters π which can minimize the error between the measured and the predicted joint torque by linear dynamic equation.

In addition to the motors lacking torque sensors, a prior calibration of the current–torque coefficient can help to obtain approximate joint torque through the measured joint current.

Figure 5 compares four sets of joint torque in the left leg. The red dotted line indicates the measured torque through the joint current. The blue solid line indicates the predicted torque through identified dynamic parameters, while the green dotted line indicates the predicted torque through identified with base dynamic parameters. The black dotted line indicates the theoretical torque calculated by parameters obtained from the 3D model. There is a large error between the measured and theoretical torque. The main reason is that the joint friction torque is not considered in the dynamics equation calculation.

Not surprisingly, the error between the measured and predicted torque is very small. Meanwhile, several torque jumps are measured when the motor changes its rotating direction. Thanks to the Coulomb friction term in our friction model, the predicted torque curve follows the measured one closely. It directly proves that the identified dynamic parameters can reflect the actual dynamic characteristics and can be used to model the physical control system.

## 6. Experimental Results and Discussion

The control approach, as mentioned above, is experimentally evaluated on the Walker3 humanoid. In addition, the balance performance is evaluated in different scenarios: push recovery on the ground, balancing on a seesaw, and push recovery on two moving skateboards. A summary of experimental videos is available at https://youtu.be/g79tWSATmhA (accessed on 31 July 2022).

### 6.1. Push Recovery on the Ground

The robot is subjected to impulses from X and Y directions while standing on the ground. A ball weighing 5 kg is used to generate impulses at the robot’s torso. The ball’s momentum with its known mass and velocity can quantify the impulses. Figure 6 shows a series of snapshots when the robot is stroked along the X axis and Y axis. Eight impulses along the X axis and seven impulses along the Y axis are exerted on the robot.

For the push recovery along the X axis, the index and magnitude of impulses are listed in Table 2.

Several key features of the tasks are drawn in Figure 7. First, it can be seen that the peak of the foot pitch angle gradually increases as the impulses increase. The foot pitch angle reaches up to 1.5${}^{\circ }$ when the impulse reaches its maximum 12 Ns. Theoretically, the foot pitch angle should be zero because the CoP constraint has been considered in the hierarchical optimization. Such a small pitch angle is acceptable given a carpet between the foot and the ground.

Figure 7b draws the CoM position along the X axis. The CoM position fluctuates in the range of −29~56 mm under the continuous impulses. Figure 7c draws the pitch angle of the torso. The robot tries to rotate the upper body to preserve the foot posture and the CoM position, which looks similar to a human rotating its trunk to maintain balance. The maximum value of the torso pitch angle is 25.7${}^{\circ }$. Increasing the impulse will cause a balance failure due to the torso pitch angle exceeding the joint angle limits.

A tiny stability error exists for these three tasks in Figure 7, although the tasks’ feedback control law works. Such a control command resulting from the tiny error will not drive the joint to move. In the video, the robot behaves in a way that it cannot recover to its original state after the disturbances.

Figure 8a compares the measured ZMP with the optimized one. The measured ZMP is calculated using the force sensor, while the optimized ZMP is calculated using the optimized foot contact force. Ideally, the measured ZMP should follow the optimized ZMP closely, but there is a slight difference. It means that all motors in the robot cannot generate the required optimized foot contact force. The main reason for that is the joint torque error due to the limited identification accuracy of the joint friction model in Section 5.2.

In addition, there are several times that the optimized ZMP reaches up to its boundary. This indicates a strong linear relationship between the optimal foot contact force, leading to the dimensionality reduction in optimal variables. As a result, the robot tends to sacrifice the lowest-priority task due to a lack of DoFs. As proof, Figure 8b plots the task error $error={∥Ax-b∥}^{2}$ of the lowest-priority task. It can be seen that the task error is small enough when the first three impulses act on the robot. However, there will always be a sharp peak with the magnitude of $10^{3}$ as long as the impulses increase to 12 Ns. The huge task error will deteriorate the control performance. The robot is originally designed to show its torso compliance according to the PD parameters, but the compliance characteristic cannot be ensured due to the task error. The inappropriate compliance will enlarge the amplitude of the torso pitch angle, which further limits the performance of push recovery. These push recovery test results show that the hQP-WBC method can handle multi-tasks well according to their priority and thus improve the robustness of the robot under environmental disturbance.

### 6.2. Balancing on a Seesaw

The robot balances the inclination disturbances along the X axis and Y axis on a seesaw. Figure 9 shows how the robot adapts to the inclined surface to maintain balance, and Figure 10 plots the measured orientation and angular velocity of the right foot. Unfortunately, no additional IMU is mounted on the seesaw to measure its real movement. Nevertheless, the estimated foot state approximates the seesaw because the ZMP resides in the foot polygon throughout the test.

The amplitude of inclined angles along both axes reaches up to 6${}^{\circ }$, as shown in Figure 10. Meanwhile, the maximum angular velocity along the Y axis is 1 rad/s, slightly larger than that on the X axis (0.6 rad/s). The reason mainly relies on the joint friction’s minor influence on the balance performance when the seesaw inclines along the Y axis. The robot needs to modulate its ankle pitch joint to adapt to the inclined seesaw along the Y axis while modulating its ankle roll joint and the length of both legs to adapt to the inclined seesaw along the X axis. Here, a low-pass filter has processed the angular velocity data with a cutoff frequency of 20 Hz.

Figure 11 shows the response of the right foot’s ZMP during the disturbances. The components of ZMP along the X and Y axes did not exceed the constrained boundary defined by foot geometry. All these prove that the task planner works well, and the robot can adjust the tasks’ target to resist the disturbance from the seesaw.

### 6.3. Push Recovery on Two Moving Skateboards

The final experiment in balance maintenance on two moving skateboards is shown in Figure 12. The two feet of the robot rest on two moving skateboards separately and suffer inclination and shift disturbances independently. The right-moving skateboard is actuated by hands to translate along the X and Y axes and rotate along the X, Y, and Z axes. At the same time, the left one is locked (Figure 12a) to evaluate the disturbance rejection capability of the robot. Figure 13 plots the measured velocity of the right foot with each direction tested separately. The shift disturbance along the Z axis is indirectly measured by rotating the skateboard about the Y axis, where the foot will rise as the tilt angle increases. The robot can resist the moving skateboard disturbance with the maximum velocities 0.94 m/s, 0.89 m/s, and 0.47 m/s along the X, Y, and Z axes, and the maximum angular velocities 1.8 rad/s, 1.4 rad/s, and 0.5 rad/s along the X, Y, and Z axes.

The robot can also maintain balance when the two skateboards have different inclination angles and translate back and forth without phase velocity. Meanwhile, when 8Ns impulses along the Y axis are exerted on the robot as the skateboards keep moving, the robot generates a large torso rotation to keep balance (Figure 12b).

The onboard computer needs to solve the hierarchical optimization, which contains four quadratic optimization problems in 1 ms, to achieve a 1 kHz control loop. Figure 14 plots the computation time of the whole algorithm, including the state estimation, trajectory planning, whole-body control, and joint-level control parts. The average and the maximum computation times are 0.363 ms and 0.639 ms, respectively. As the most time-consuming portion, the computation time of the whole-body control part is also plotted here in the yellow line. The average computation time is 0.321 ms, which takes up about 88 percent of the time, leaving 0.042 ms for the other parts. We must solve the quadratic optimization and the null-space projection matrix sequentially in the whole-body control part. About 72% of the computation time is used for quadratic optimization, and the left 28% is used for the null-space projection matrix. The quadratic optimization is solved through a C++ open-source QP solver, qpOASES [25], which implements the active set algorithm. The null-space projection matrix must calculate the pseudo-inverse of the matrix first, and the complete orthogonal decomposition algorithm implemented in the Eigen matrix library is used here.

## 7. Conclusions

This paper aims to make the proprioception-actuated humanoid robot capable of dynamic balance. For this purpose, tasks and constraints are assigned in a hierarchy of task priorities.

A real-time computation is achieved through computationally efficient WBC software and a reduced hierarchical whole-body control scheme.UBTMaster, a modular control system with real-time communication and powerful computing capabilities, is designed.The key dynamic parameters are identified to deal with the nonlinear friction and imprecision of the model of the robot.

Results show that the predicted torque is close to the measured, and the average residual is less than 1 Nm.

After fully considering these aspects in a system, the balance performance of the humanoid robot Walker3 is tested in various scenarios. The robot can be balanced with continuous impulses on the X and Y axes up to 12 Ns. Like human behavior, the robot tends to rotate its upper body to maintain foot posture and CoM position. In addition, we found reasons to limit push recovery performance. When the optimal variable reaches the constraint boundary, dimensionality reduction will cause the system to sacrifice the task with the lowest priority. Because of the strict hierarchy, high-priority tasks are not affected. An effective solution is to provide more redundant degrees of freedom, such as adding arms to the robot.

When Walker3 stands on the seesaw, it can actively adapt to the tilted surface to maintain balance. Different from [15], the state of the seesaw is estimated without an additional IMU and then used to update the trajectory of the task. The experimental results show that the inclination angle along the two axes reaches 6${}^{\circ }$. Meanwhile, the maximum angular velocities of the X axis and Y axis are 0.6 rad/s and 1 rad/s, respectively, which is about 1.7~2.8 times the performance of the torque control robot COMAN [26].

To further exploit the robot’s adaptability to uncertain perturbations, we placed Walker3 on two moving skateboards and applied tilt and displacement perturbations. The maximum velocities of the robot in the X, Y, and Z axes are 0.94 m/s, 0.89 m/s, and 0.47 m/s, respectively. The maximum angular velocities of the X, Y, and Z axes are 1.8 rad/s, 1.4 rad/s, and 0.5 rad/s, respectively. When the two skateboards have different inclination angles and no phase velocity in front and back translation, the robot can even resist 8 Ns impulse.

The results show that the proprioception–actuation robot can perform quite well as the torque-controlled robot under a strict hierarchical structure, real-time calculation, and careful joint friction treatment. We hope to improve the balancing framework through intelligent planning to cope with the greater disruption of future work. Furthermore, the results show that the introduction of online predictive control technology will significantly improve the robustness of the robot.

At present, our research focuses on the effectiveness of the proposed method. Our research goal is to focus on the solution of biped robot dynamic balance problem from the perspective of algorithm, software, and hardware, and the adopted Walker is used to test this solution. In terms of the performance of the robot, Walker robot needs to increase its arms to improve its balance performance more fully, which is comparable to, or goes beyond, the Atlas robot.

## Figures and Tables

**Figure 1 micromachines-13-01458-f001:**
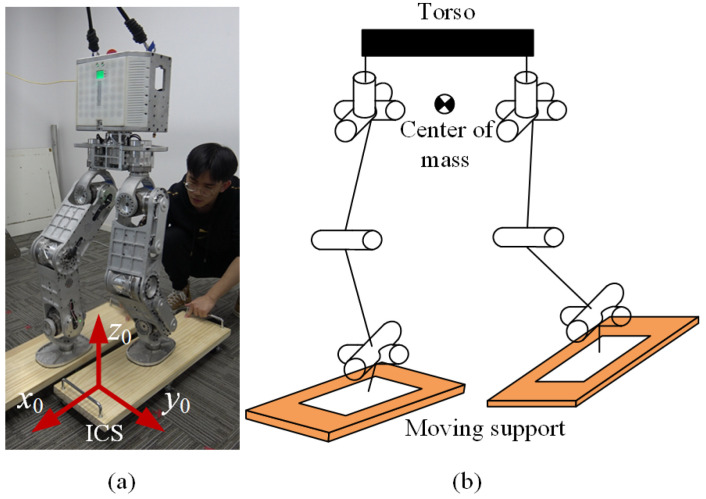
Walker3 humanoid robot with 12 actuated DoFs (**a**) and its kinematic model (**b**).

**Figure 2 micromachines-13-01458-f002:**
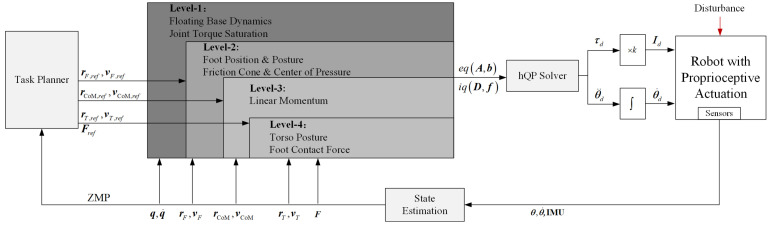
Overview of the control architecture.

**Figure 3 micromachines-13-01458-f003:**
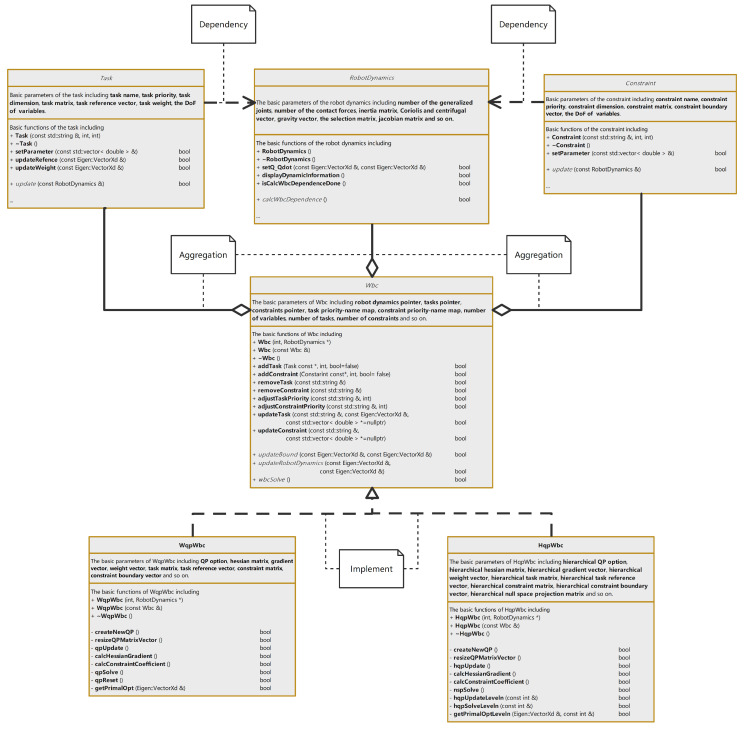
The architecture of WBC software.

**Figure 4 micromachines-13-01458-f004:**
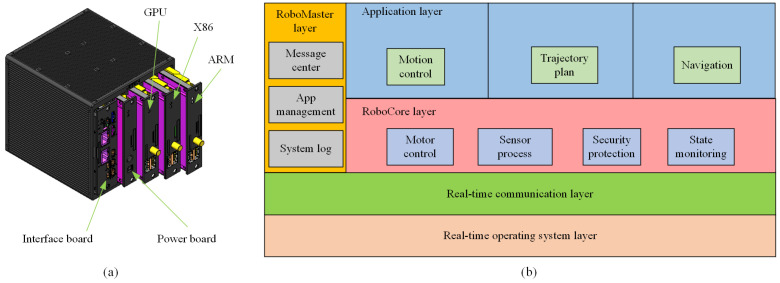
The modular master control system (**a**) and its software architecture (**b**).

**Figure 5 micromachines-13-01458-f005:**
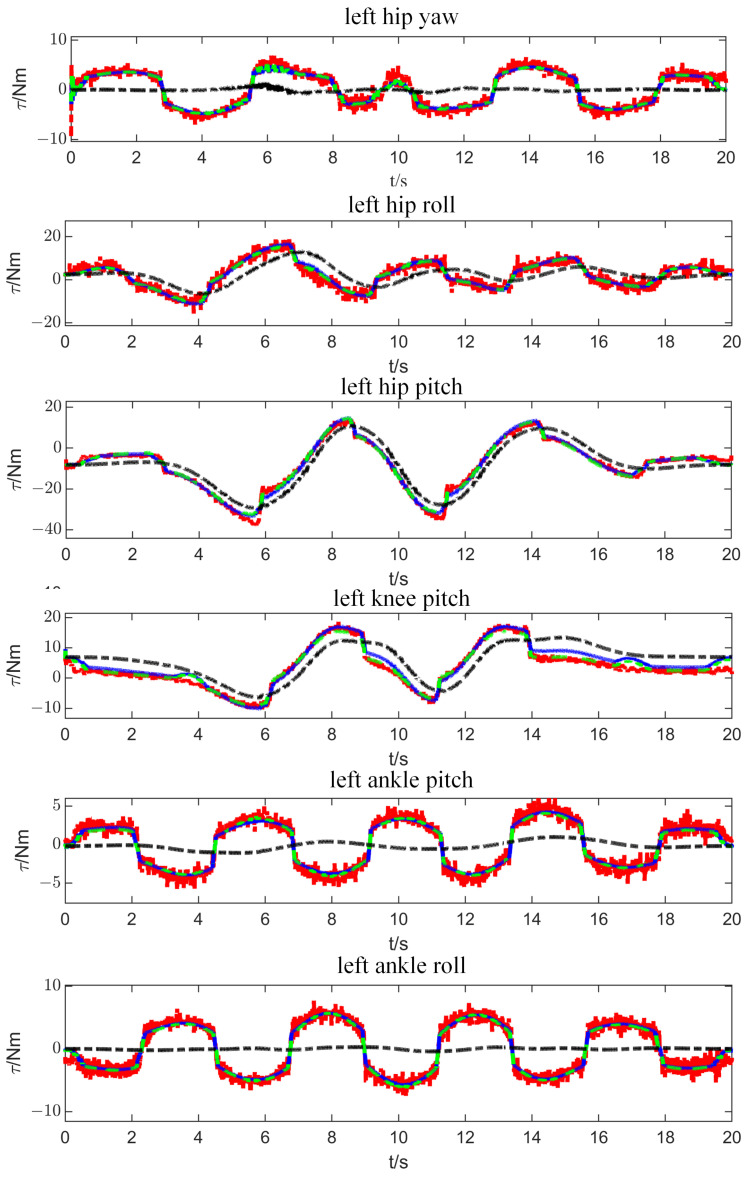
The comparison of joint torque in the left leg.

**Figure 6 micromachines-13-01458-f006:**
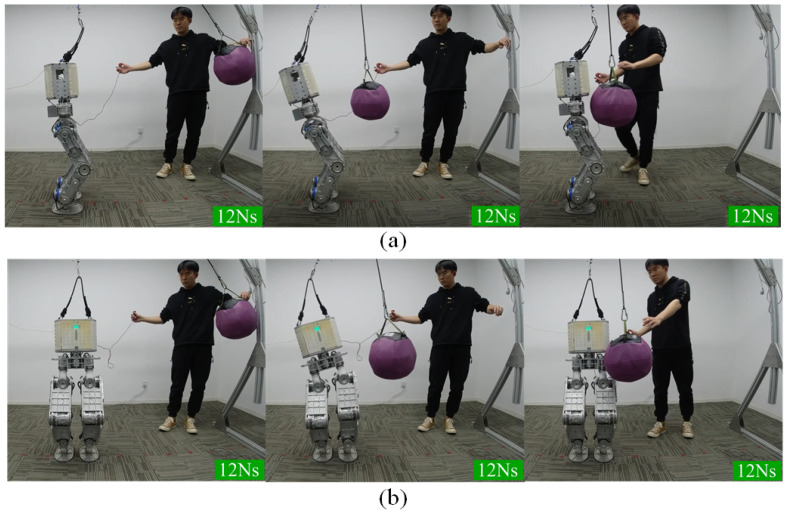
The balancing behavior in push recovery scenario along the X axis (**a**) and the Y axis (**b**).

**Figure 7 micromachines-13-01458-f007:**
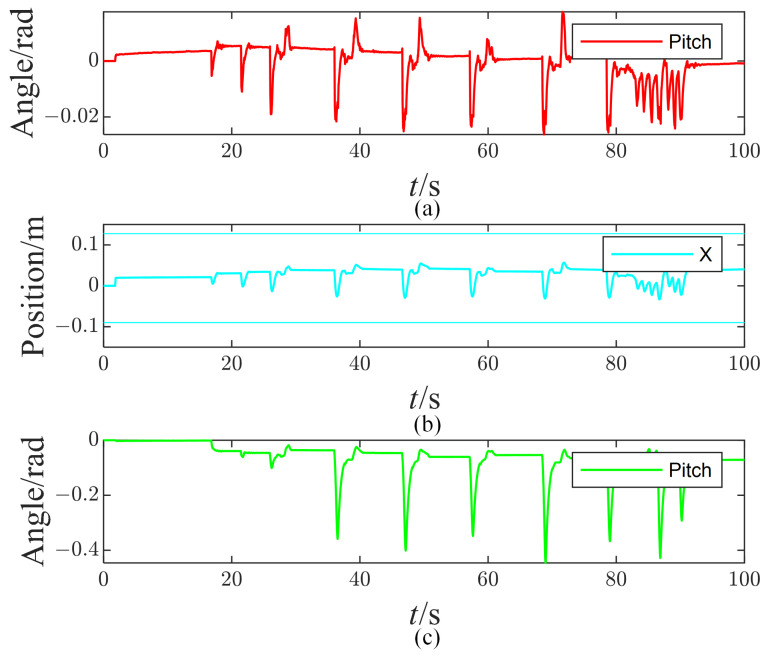
The measured pitch angle of the right foot (**a**), CoM position along the X axis (**b**), and the pitch angle of the torso (**c**).

**Figure 8 micromachines-13-01458-f008:**
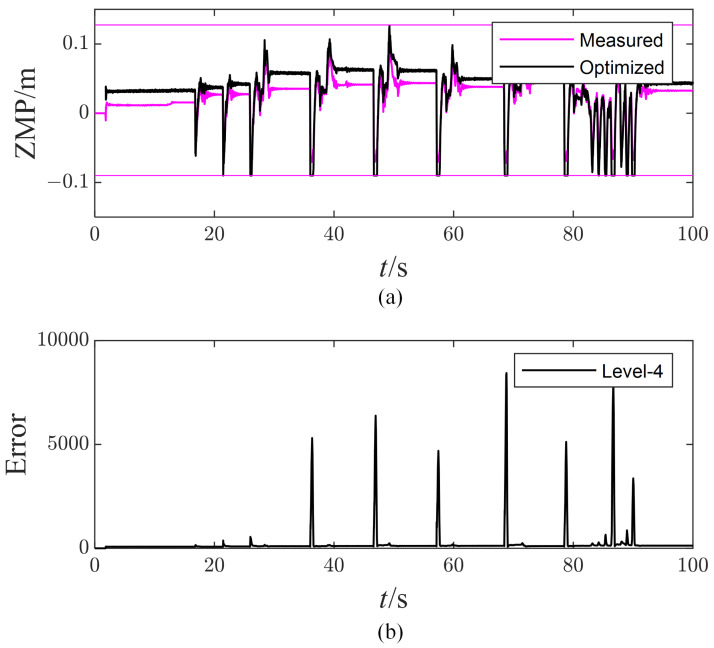
The measured and optimized ZMP of right foot (**a**) and the task error of the lowest priority task (**b**).

**Figure 9 micromachines-13-01458-f009:**
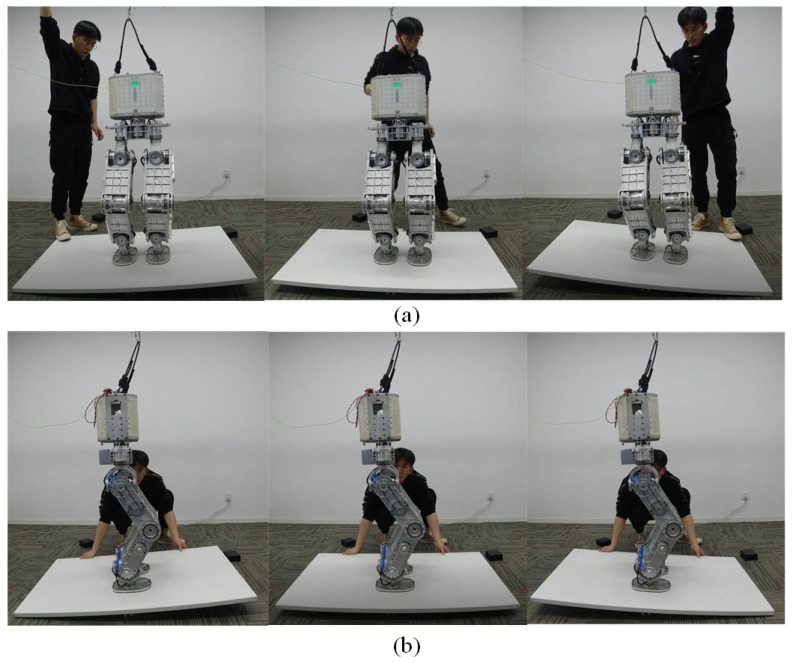
The balancing behaviors when the seesaw rotates along the X axis (**a**) and the Y axis (**b**).

**Figure 10 micromachines-13-01458-f010:**
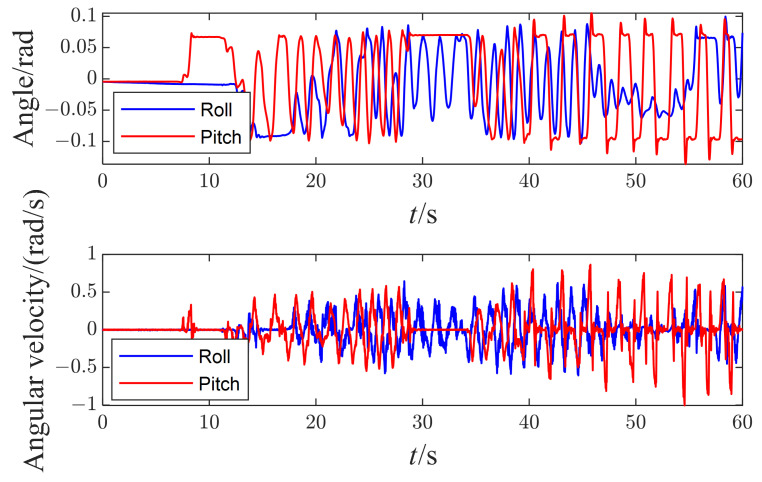
The measured orientation and angular velocity of the right foot.

**Figure 11 micromachines-13-01458-f011:**
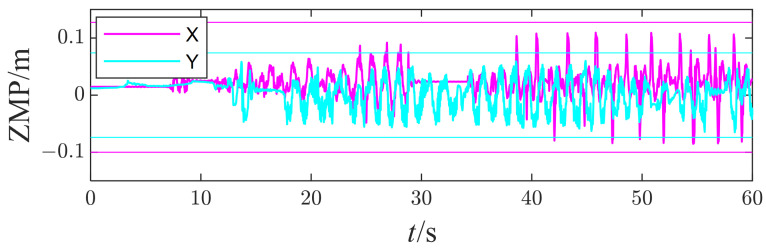
The measured ZMP of the right foot.

**Figure 12 micromachines-13-01458-f012:**
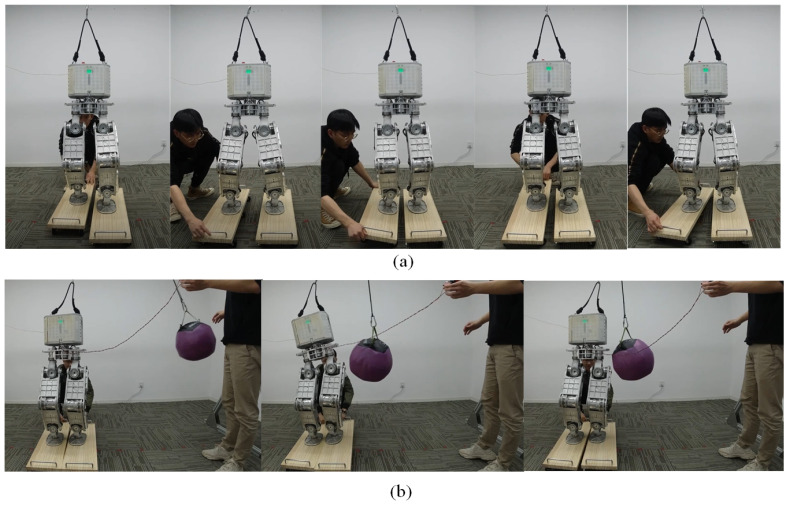
The balancing behaviors when the right support moves in all directions (**a**) and the balancing behaviors in push recovery on the moving support scenario (**b**).

**Figure 13 micromachines-13-01458-f013:**
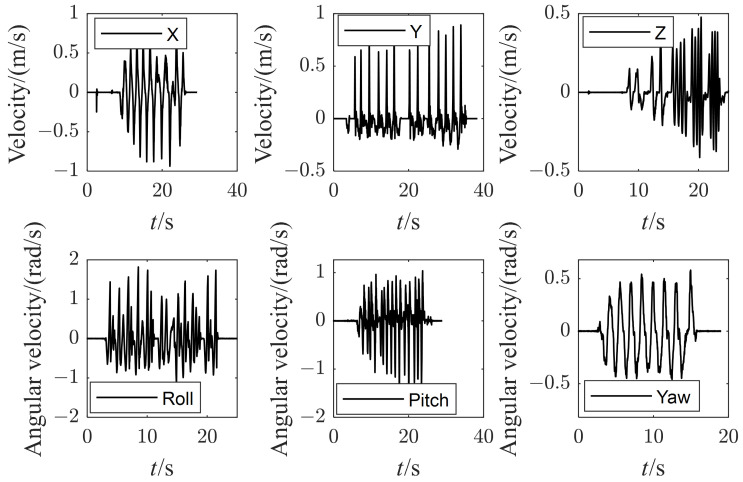
The measured velocity of the right foot with each direction.

**Figure 14 micromachines-13-01458-f014:**
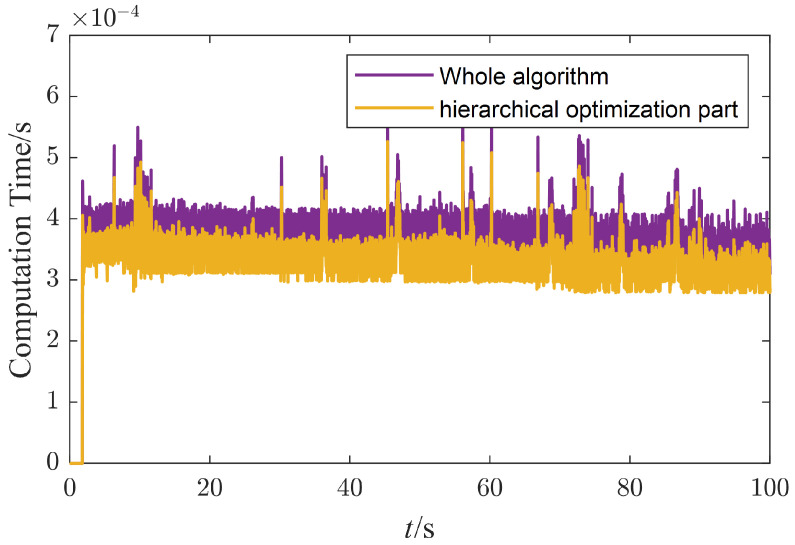
The computation time of the algorithm.

**Table 1 micromachines-13-01458-t001:** Table task constraint hierarchy.

Level	Task	Task Dimensions	Constraint	Constraint Dimensions
1	Floating base dynamics	6	Joint torque saturation	12
2	Foot position and posture	12	Center of pressure and Friction cone	18
3	Linear momentum	3		
4	Torso posture and Foot contact force	15		

**Table 2 micromachines-13-01458-t002:** The magnitude of impulses along the X axis.

Index	1	2	3	4∼8
Magnitude	8 Ns	10 Ns	11 Ns	12 Ns

## Data Availability

Not applicable.

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
