# Peer review of "Dynamic Balancing of Humanoid Robot with Proprioceptive Actuation: Systematic Design of Algorithm, Software, and Hardware"

_micromachines, 2022, doi:10.3390/mi13091458_

Round 1

Reviewer 1 Report

Interesting paper in subject of Micromachines journal of MDPI.

Article need minor review for improve of description of scientific background of article and readability. Also need to improvesome mistakes.

My remarks was presented in some comments. Please consider it in your review.

Comment 1

Article was based on restricted number of references (23). It need to add some article which show new possibilities of bipedal gait of robots. Please to found some actual references from last year (2018-2022). I propose add

Recent Advances in Bipedal Walking Robots: Review of Gait, Drive, Sensors and Control Systems. Sensors 202222, 4440. https://doi.org/10.3390/s22124440

Bipedal Humanoid Hardware Design: a Technology Review. Curr Robot Rep 2, 201–210 (2021). https://doi.org/10.1007/s43154-021-00050-9

Prototype Model of Walking Robot. AMM 2014;613:21–8. https://doi.org/10.4028/www.scientific.net/amm.613.21

Comment 2

In Line 39 you wrote

Dietrich et al.[ 3 ] have categorized the existing WBC methods into (1) null space projection- 39

based WBC (NSP-WBC), (2) weighted quadratic program-based WBC (WQP-WBC), and (3) 40

hierarchical quadratic program based WBC (HQP-WBC).

I propose to write

Dietrich et al.[ 3 ] have categorized the existing WBC methods into: null space projection- based WBC (NSP-WBC), weighted quadratic program-based WBC (WQP-WBC), and hierarchical quadratic program based WBC (HQP-WBC).

Or eventually use of bullet

Comment 3

In Line 49 you wrote

This method has been applied to the Atlas robot to execute multi tasks during the DARPA Robotics Challenge [ 6 ] [ 7 ] [ 8 ].

I propose to write in this mode

This method has been applied to the Atlas robot to execute multi tasks during the DARPA Robotics Challenge [ 6- 8 ].

Comment 4

You wrote section 6 as

6. Experimental validation

Because you present the experimental results and also present its discussion I propose to change on

6. Experimental validation and discussion

Or

6. Experimental resu;ts and Discussion

Comment  5

Figure 13 and 14 need move from Conclussion to previous section

Comment 6

I propose use bullet in presentation of main achievements of article

Comment 7

Before references I propose add

References

Comment 8

I propose add to article Abreviations 

Author Response

We would like to thank all the Reviewers for their careful reading of the manuscript and for raising important points in their assessment. We strongly believe that their high-quality review and constructive comments have led to a greatly improved quality of our revised paper. We thank all the Reviewers for their valuable time in reviewing this paper, and we hope our substantial efforts to address all reviewers’ concerns will be recognized. The manuscript has been revised according to the valuable comments. Please see the attachment for our reply to reviewer.

Reviewer 2 Report

Please check attached file.

Author Response

(The authors gave the same response as above.)

Round 2

Reviewer 2 Report

Thanks for your reply.